

# Position operators and interband matrix elements of scalar and vector potentials in the 8-band Kane model

Ivan A. Ado[1], M. Titov[1], Rembert A. Duine[2,3] and Arne Brataas[4]

**1** Radboud University, Institute for Molecules and Materials,
Heyendaalseweg 135, 6525 AJ Nijmegen, The Netherlands
**2** Institute for Theoretical Physics, Utrecht University,
Princetonplein 5, 3584 CC Utrecht, The Netherlands
**3** Department of Applied Physics, Eindhoven University of Technology,
P.O. Box 513, 5600 MB Eindhoven, The Netherlands
**4** Center for Quantum Spintronics, Department of Physics,
Norwegian University of Science and Technology,
Høgskoleringen 5, 7491 Trondheim, Norway

⋆ iv.a.ado@gmail.com

## Abstract

We diagonalize the 8-band Kane Hamiltonian with a proper inclusion of the interband matrix elements of the scalar and vector potentials. This leads, among other results, to a modification of the conventional expression for the spin-orbit coupling (SOC) strength in narrow-gap semiconductors with the zinc blende symmetry. We find that in GaAs, at low temperatures, the correct expression for the SOC strength is actually twice as large as usually considered. In InSb it is 1.76 times larger. We also provide a proper treatment of the interband matrix elements of the position operator. We show that the velocity operator in a crystal should be defined as a time-derivative of a fictitious position operator rather than the physical one. We compute the expressions for both these position operators projected to the conduction band of the 8-band Kane model. We also derive an expression for the projected velocity operator and demonstrate that the SOC strength in it differs from the SOC strength in the Hamiltonian. The ratio between them is not equal to 1, as it is often assumed for the Rashba model. It does not equal 2 either. The correct result for this ratio is given by a rational function of the parameters of the model. This function takes values between $4(23 + 3\sqrt{2})/73 \approx 1.49$ and 2. Our findings modify a vast number of research results obtained using the Rashba model and provide a path for consistent treatment of the latter in future applications.

## 1  Introduction

Using low-energy models that take into account just a few most relevant bands of a solid is often sufficient to accurately describe transport, optical properties, and strain effects [1, 2]. Essential features of conceptually nontrivial systems such as topological insulators, are, in principle, captured [3] by the 6-band Luttinger-Kohn Hamiltonian [4] and generalizations [5] of the 8-band Kane model [6]. In graphene, only two bands are retained from the full band structure [7].

One of the most widely used approaches to derive low-energy models for solids is the $\boldsymbol{k} \cdot \boldsymbol{p}$ method [2,4,6,8,9] supplemented with the envelope function approximation (EFA) [2,4,10–13]. The latter enables the inclusion of effects of external electric and magnetic fields that break the translational symmetry of the crystal. The Luttinger-Kohn Hamiltonian, the 8-band Kane model, and different extended versions of the latter are all obtained with this approach.

In the $\mathbf{k} \cdot \mathbf{p}$ paradigm, bands of a solid are coupled for finite values of the quasimomentum $\mathbf{k}$.[1] It is often convenient to decouple them from each other by a transformation that eliminates the corresponding interband matrix elements of the Hamiltonian. In the 8-band Kane model, this is done in order to derive an effective Hamiltonian for the conduction band. The three closest valence bands of the same model can also be described by a similarly "decoupled" Hamiltonian, but one needs to additionally take into account effects of some distant bands.[2] This actually results in the Luttinger-Kohn Hamiltonian.

Such a decoupling of bands in solids is, in fact, very similar [14] to the decoupling of electrons and positrons in the relativistic theory that one performs to derive the Pauli Hamiltonian from the Dirac Hamiltonian in vacuum. Based on this similarity, analogs of fully relativistic effects, such as the Zitterbewegung, have been predicted to occur also in semiconductors [15]. Likewise, spin-orbit coupling (SOC) in narrow-gap semiconductors emerges in the same way as it does in vacuum when one removes positrons by a unitary transformation. [14].

There exists however an important difference between the decoupling of bands in crystals and in the Dirac theory. In the latter, the spinor basis consists of four elements: two for electrons, and two for positrons. To project an operator or a state to an element of this basis, only the matrix product operation is required. In a crystal this is not the case because projection to a particular band includes a real space integration over the unit cell. As a result of the integration, position-dependent operators, e.g. the position operator itself, acquire finite interband matrix elements (dipole moments) [16, 17]. In the Dirac theory, they do not.

One of the spatial-dependent objects with nonzero interband matrix elements of this origin is the scalar potential $V(\mathbf{r})$. If we consider its matrix element between two cell-periodic functions of the $s$ and $p$ atomic symmetry respectively, we will see that such an element is finite and, moreover, that it is linearly dependent on the gradient of $V(\mathbf{r})$, e. g.

$$\langle S \mid V \mid X \rangle = (\partial V / \partial x) \langle S \mid x \mid X \rangle \neq 0 . \tag{1}$$

Here the scalar product $\langle \ldots \rangle$ denotes an integral over the unit cell. In Eq. (1), we have disregarded higher derivatives of the potential. Thus, the first derivative can be computed at any point within the cell. The gradient $\partial V / \partial \mathbf{r} \equiv \boldsymbol{\nabla} V$ is a very important quantity, for example in systems with structural inversion asymmetry, in which it determines the effective SOC of the Rashba type [14, 18]. The latter in turn contributes to the anomalous Hall effect, the spin Hall effect, and other effects originating from SOC [19].

In the 8-band Kane model, the effective SOC contribution to the Hamiltonian is proportional to the gradient $\boldsymbol{\nabla} V$ of the scalar potential and is commonly considered to originate solely from its intraband matrix elements [20]. At the same time, the interband matrix elements of $V$ that, according to Eq. (1), are also proportional to $\boldsymbol{\nabla} V$ are completely ignored in most treatments of the 8-band Kane model (see, e.g. Refs. [2, 14, 21]). In a recent work [22], effects of the interband matrix elements on the effective SOC were studied using first principles for valence bands in GaAs. Here, we develop an analytical theory for the effective SOC that includes contributions from both the intraband and the interband matrix elements of the scalar potential. We use the 8-band Kane model as an example, but the idea is general and extends to other models. In what follows, we refer to our example model as "the Kane model".

We also derive an expression for the velocity operator $\mathbf{v}^{(c)}$ of the conduction (c) band in the Kane model, taking into account the interband matrix elements of the scalar potential in addition to the usually considered intraband ones. We compare the SOC-induced contribution to $\mathbf{v}^{(c)}$ that is proportional to $\boldsymbol{\nabla} V$ with the similar contribution to the Hamiltonian. It turns out that, because of the interband matrix elements of the scalar potential, these two contributions

---

[1]Or the envelope functions momentum.

[2]Otherwise the effective mass for the heavy-hole band has an incorrect sign.

are determined by two different SOC strengths.[3] Thus, for example, in the Rashba model [23] one cannot quantify the Rashba contributions to the Hamiltonian and to the velocity operator by a single constant $\alpha_{\mathrm{R}}$.

Moreover, as we demonstrate in this paper, one cannot even define the velocity operator as a commutator of the physical position operator and the Hamiltonian, once the full band structure has been replaced by a finite number of bands. There is however a workaround to this issue, at least in certain cases. By defining a fictitious position operator $\mathbf{r}$ as the one that forms a canonical pair with the operator $\mathbf{k}$ in EFA, one can still define velocity by commuting $\mathbf{r}$ with the Hamiltonian. This allows, in particular, to project the Kubo formula for the dc conductivity tensor to single bands [24]. The need for two different position operators will be explained later in the text with more details.

Another object, interband matrix elements of which are usually ignored, is the vector potential. We take these matrix elements into account and find that they affect the magnetic field dependent renormalization of the effective mass. The effective g-factor, however, remains intact.

The analysis of the Kane model that we perform in this work is most easily formulated in terms of operators in Hilbert space and unitary transformations applied to them. Such an operator-type formulation for EFA is still missing in the literature, and in this work we take the opportunity to present it. We note that effective Hamiltonians are often obtained from symmetry considerations, and operators of other observables are then derived from these Hamiltonians. In some cases this approach gives incorrect results [24]. We argue that treating all observables on equal footing and applying the same operator transformations to all of them is preferable. This paper, in particular, demonstrates how this is achieved in EFA.

The paper is organized as follows. In Sec. 2 we describe EFA and the Kane model. In addition to the common ingredients of the latter, we consider the interband matrix elements of the scalar and vector potentials. In Sec. 2.3, we demonstrate how they are computed in the 8-band basis. Secs. 3.1, 3.2 present a general (perturbative) scheme that can be used to diagonalize Hamiltonians with a block structure. The scheme employs the language of unitary transformations and operator identities. We also show how the observables other than the Hamiltonian are transformed in it. In Sec. 3.3, the diagonalization scheme is applied to the Kane model with a focus on its conduction band. Secs. 4.1, 4.2 introduce the concept of the fictitious position that should be used in crystals, instead of the physical position, to define the velocity operator. We derive the expressions for the velocity operator and both position operators in the conduction band of the diagonalized Kane model in Sec. 4.3. We compare different SOC strengths that enter the expressions for all the computed operators in Table 2. The role of the fictitious position operator in the Kubo formula for dc conductivity is explained in Sec.4.4. We comment on the correct way of defining the velocity operator in the Rashba model in Sec. 4.5. Possible connection with experiments is discussed in Sec. 4.6

## 2 Kane model in the envelope function approximation

### 2.1 8 bands and the envelope function approximation

The Kane model [6] describes materials with the zinc blende symmetry (e.g., GaAs, InSb) by considering a single conduction band and three valence bands. Together with the spin degree of freedom, this gives 8 bands. Each of them is determined by a solution of a Schrödinger

---

[3]The ratio between them is a rational function of the parameters of the model. It takes the value of 2 only in the limit of vanishing band gap.

equation for cell-periodic functions,

$$(-\hbar^2 \boldsymbol{\nabla}^2/2m + \mathcal{W})U_i = E_i U_i\,, \tag{2}$$

where $\mathcal{W}$ is the atomic periodic potential. We denote the 8 relevant solutions $U_i$ by $S_\uparrow, S_\downarrow, X_\uparrow$, $X_\downarrow, Y_\uparrow, Y_\downarrow, Z_\uparrow, Z_\downarrow$. The first two of them have the atomic $s$ symmetry and describe the conduction band. The remaining six are of the $p$ type and refer to the valence bands. The subscripts $\uparrow, \downarrow$ correspond to spin up and spin down states, respectively. Each function $U_i = U_i(\boldsymbol{r}, \sigma)$ is a two-component spinor with the spatial periodicity of the crystal, and $\sigma$ describes the spin degree of freedom. In the presence of slowly varying external fields, it is assumed that each electron state is expressed as

$$\Psi(\boldsymbol{r}, \sigma) = \sum_i \psi_i(\boldsymbol{r}) U_i(\boldsymbol{r}, \sigma)\,, \tag{3}$$

where $\psi_i(\boldsymbol{r})$ are scalar envelope functions considered constant within each unit cell of the crystal. This is the basis of EFA.

It is clear from Eq. (3) that a state $\Psi$ is fully determined by a vector with the components $\psi_i(\boldsymbol{r})$, which thus provides the EFA-representation of $\Psi$. For a given $\Psi$, the component $\psi_i(\boldsymbol{r})$ is uniquely defined as a band projection:

$$\psi_i(\boldsymbol{r}) = \int\limits_{\substack{\text{unit cell} \\ \text{containing } \boldsymbol{r}}} d^3 r' \sum_{\sigma=\uparrow,\downarrow} U_i^*(\boldsymbol{r}', \sigma) \Psi(\boldsymbol{r}', \sigma) \equiv \langle U_i | \Psi \rangle(\boldsymbol{r})\,, \tag{4}$$

where we use Eq. (3) and the fact that envelope functions do not vary within the cell. The scalar product $\langle \dots \rangle(\boldsymbol{r})$ in Eq. (4) is defined as an integral over the unit cell that contains $\boldsymbol{r}$ augmented with a sum in the spin space. The argument of $\langle \dots \rangle(\boldsymbol{r})$ refers to this particular unit cell.

Now let us consider the action of an arbitrary operator $F$ on $\Psi$. To express it in EFA at the position $\boldsymbol{r}$, one should project it to the $i$-th band in the unit cell that contains $\boldsymbol{r}$. By analogy with Eq. (4), we can write:

$$(F\Psi)_i(\boldsymbol{r}) = \int\limits_{\substack{\text{unit cell} \\ \text{containing } \boldsymbol{r}}} d^3 r' \sum_{\sigma=\uparrow,\downarrow} U_i^*(\boldsymbol{r}', \sigma) F\Psi(\boldsymbol{r}', \sigma) = \sum_j \langle U_i|F|U_j \rangle(\boldsymbol{r}) \psi_j(\boldsymbol{r})\,. \tag{5}$$

Thus, in EFA, $F$ is represented by a matrix with the operator valued elements $F_{ij} = \langle U_i|F|U_j \rangle$. This matrix acts on a vector with the components $\psi_j$. For brevity, we omit the spatial arguments of states and operators in EFA here and below.

## 2.2 Kane Hamiltonian: Part I

The Kane Hamiltonian is the EFA representation of the Hamiltonian of Eq. (2) in the presence of SOC and slowly varying scalar potential $V$ and vector potential $\boldsymbol{A}$. The basis for the representation is provided by the 8 functions $U_i$, and SOC is taken into account perturbatively by means of the matrix elements in this basis.

In order to derive the Kane Hamiltonian, we first need to expand $V$ and $\boldsymbol{A}$ inside each unit cell. For this, we introduce a notation slightly different from that of Eq. (1). For a particular unit cell $\mathcal{C}$ and a fixed position $\boldsymbol{r}$ inside it, we write

$$V(\boldsymbol{r} + \boldsymbol{\rho}) = V(\boldsymbol{r}) + \boldsymbol{\rho} \boldsymbol{\nabla} V\,, \tag{6}$$

where $\boldsymbol{\rho}$ is a vector lying in the unit cell that contains the origin $\boldsymbol{r} = 0$. In this work, we neglect second (and higher) derivatives of both $V$ and $A$, hence $\boldsymbol{r}$ in the cell $\mathcal{C}$ can be chosen arbitrary. For example it can be at the cell center. For the vector potential, we write similarly, employing the symmetric gauge,[4]

$$A(\boldsymbol{r} + \boldsymbol{\rho}) = A(\boldsymbol{r}) + \frac{1}{2}[\boldsymbol{B} \times \boldsymbol{\rho}], \tag{7}$$

where $\boldsymbol{B} = \boldsymbol{\nabla} \times \boldsymbol{A}$ is the external magnetic field. Consistent with the assumption above, we will disregard all spatial derivatives of $\boldsymbol{B}$ throughout the paper. From now on, we will also omit the argument in both $V(\boldsymbol{r})$ and $A(\boldsymbol{r})$.

Following the standard convention, we introduce the operator $\boldsymbol{p} = -i\hbar\boldsymbol{\nabla}$ that acts solely on the cell functions $U_i$, and the operator $\hbar\boldsymbol{k} = -i\hbar\boldsymbol{\nabla} - e\boldsymbol{A}/c$ that acts only on the envelope functions $\psi_i$. Using these notations, one can express the total Hamiltonian as

$$H = \frac{1}{2m}\left(\boldsymbol{p} + \hbar\boldsymbol{k} - \frac{e}{2c}[\boldsymbol{B} \times \boldsymbol{\rho}]\right)^2 + \mathcal{W} + \frac{\hbar}{4m^2c^2}[\boldsymbol{\nabla}\mathcal{W} \times \boldsymbol{p}]\boldsymbol{\sigma} + V + \boldsymbol{\rho}\boldsymbol{\nabla}V + \mu_B\boldsymbol{\sigma}\boldsymbol{B}_{\text{eff}}, \tag{8}$$

where $\mu_B = |e|\hbar/2mc$ is the Bohr magneton, $\boldsymbol{\sigma}$ is the vector of Pauli matrices, $\boldsymbol{B}_{\text{eff}} = \boldsymbol{B} + \boldsymbol{B}_{\text{exc}}$ is the effective magnetic field, and $\boldsymbol{B}_{\text{exc}}$ is the constant exchange field that we added for completeness. $\boldsymbol{B}_{\text{exc}}$ can originate, e.g. from doping of the material by magnetic impurities. For simplicity, we assume that the electron $g$-factor equals 2. The SOC term in Eq. (8) largely exceeds both the similar $\boldsymbol{p}$-dependent contribution from $\boldsymbol{\nabla}V$ and the vacuum $\boldsymbol{k}$-dependent contribution from $\boldsymbol{\nabla}\mathcal{W}$, which are therefore disregarded. The $\boldsymbol{k}$-dependent vacuum SOC contribution from $\boldsymbol{\nabla}V$ is omitted too because the $\boldsymbol{k}$-dependent effective SOC (that is also proporional to $\boldsymbol{\nabla}V$) is usually much stronger.[5] In GaAs, it is $10^6$ times stronger [25, 26], in InSb $10^8$ times [27].

The EFA representation of $H$ is an operator valued matrix with the elements $H_{ij} = \langle U_i|H|U_j\rangle$. In order to compute them, we separate all terms in Eq. (8) into three groups,

$$H = \underbrace{\frac{\boldsymbol{p}^2}{2m} + \mathcal{W} + \frac{\hbar}{4m^2c^2}[\boldsymbol{\nabla}\mathcal{W} \times \boldsymbol{p}]\boldsymbol{\sigma}}_{\text{cell-periodic}} + \underbrace{\frac{\hbar^2\boldsymbol{k}^2}{2m} + V + \mu_B\boldsymbol{\sigma}\boldsymbol{B}_{\text{eff}}}_{\text{envelopes}} + \underbrace{\frac{\hbar}{m}\boldsymbol{k}\boldsymbol{p} + \boldsymbol{\rho}\boldsymbol{\nabla}V - \frac{e\hbar}{2mc}[\boldsymbol{k} \times \boldsymbol{B}]\boldsymbol{\rho}}_{\text{"k dot p"}}, \tag{9}$$

where we ignored the "quadrupole-like" terms $\propto \rho_i\rho_j$ and $\propto p_i\rho_j$. The matrix elements $H_{ij}$ of the terms in the first and in the second groups are nonzero only between either the $s$ functions or the $p$ functions. For the first group, these elements are just constant values. For the second, they are represented by operators that act on the envelope functions. The third group gives finite matrix elements between cell functions of different symmetry. Let us explain how they are computed.

## 2.3  k dot p elements

The $\boldsymbol{kp}$ term in Eq. (9) is commonly expressed in EFA using the real-valued matrix elements

$$P = \frac{\hbar}{im}\langle S_{\uparrow,\downarrow}|p_x|X_{\uparrow,\downarrow}\rangle = \frac{\hbar}{im}\langle S_{\uparrow,\downarrow}|p_y|Y_{\uparrow,\downarrow}\rangle = \frac{\hbar}{im}\langle S_{\uparrow,\downarrow}|p_z|Z_{\uparrow,\downarrow}\rangle. \tag{10}$$

The other two terms of the "k dot p" group in Eq. (9) depend on the coordinate $\boldsymbol{\rho}$ rather than on the momentum. In our case, however, they can also be expressed by means of the matrix elements of Eq. (10), as long as the system can be assumed infinite. This is achieved with the help of the so-called $p - r$ relation [28], which we will now implement.

---

[4]In which the vector potential for the constant magnetic field is expressed as $A(\boldsymbol{r}) = [\boldsymbol{B} \times \boldsymbol{r}]/2$.

[5]The omitted SOC terms: $(\lambda_{\text{vac}}/\hbar)[\boldsymbol{\nabla}V \times \boldsymbol{p}]\boldsymbol{\sigma}$, $\lambda_{\text{vac}}[\boldsymbol{\nabla}\mathcal{W} \times \boldsymbol{k}]\boldsymbol{\sigma}$, and $\lambda_{\text{vac}}[\boldsymbol{\nabla}V \times \boldsymbol{k}]\boldsymbol{\sigma}$, where $\lambda_{\text{vac}} = \hbar^2/4m^2c^2$.

We know that $[p_j, \rho_l] = -i\hbar\delta_{jl}$, where $\delta$ is the Kronecker delta. From this, we immediately derive $[\boldsymbol{p}^2, \rho_l] = -2i\hbar p_l$. Using the fact that $(\boldsymbol{p}^2/2m + \mathcal{W})U_i = E_i U_i$, one can find, e.g.

$$\langle S_{\uparrow,\downarrow}|\rho_l|X_{\uparrow,\downarrow}\rangle = \frac{\langle S_{\uparrow,\downarrow}|[\boldsymbol{p}^2, \rho_l]|X_{\uparrow,\downarrow}\rangle}{2m(E_s - E_p)} = \frac{1}{E_s - E_p}\frac{\hbar}{im}\langle S_{\uparrow,\downarrow}|p_l|X_{\uparrow,\downarrow}\rangle = \frac{P}{\xi}\delta_{xl}, \tag{11}$$

with $\xi = E_s - E_p > 0$. For $Y_{\uparrow,\downarrow}$ and $Z_{\uparrow,\downarrow}$, the procedure is similar.

In practice the function $iS_{\uparrow,\downarrow}$ is considered instead of $S_{\uparrow,\downarrow}$. Using Eqs. (10), (11), it is straightforward to observe that the entire "k dot p" group in Eq. (9) (the last three terms on the right-hand side of it) produces the following matrix element between $iS_{\uparrow,\downarrow}$ and $X_{\uparrow,\downarrow}$:

$$\left\langle iS_{\uparrow,\downarrow}\left|\frac{\hbar}{m}\boldsymbol{k}\boldsymbol{p} + \rho\boldsymbol{\nabla}V - \frac{e\hbar}{2mc}[\boldsymbol{k}\times\boldsymbol{B}]\rho\right|X_{\uparrow,\downarrow}\right\rangle = P\widetilde{k}_x, \tag{12}$$

where we introduced the notation

$$\widetilde{\boldsymbol{k}} = \boldsymbol{k} - \frac{i}{\xi}\boldsymbol{\nabla}V - \frac{i\mu_{\mathrm{B}}}{\xi}[\boldsymbol{k}\times\boldsymbol{B}]. \tag{13}$$

When $Y_{\uparrow,\downarrow}$ and $Z_{\uparrow,\downarrow}$ are considered instead of $X_{\uparrow,\downarrow}$ in Eq. (12), the result is the same with $\widetilde{k}_x$ replaced by $\widetilde{k}_y$ and $\widetilde{k}_z$, respectively.

As we see, $\widetilde{k}_i$ is different from $k_i$ due to the interband matrix elements of the scalar and vector potentials. It is this difference that leads, in particular, to a modification of the standard expression for the effective SOC strength in the conduction band of the Kane model, and thus also in the Rashba model.

## 2.4 Kane Hamiltonian: Part II

In the Kane model, SOC is taken into account by means of the matrix elements of the third term (of the first group) in Eq. (9) between the $p$ functions. They are all expressed with the help of a single parameter

$$\Delta = \frac{3i\hbar}{4m^2c^2}\left\langle X_{\uparrow,\downarrow}\left|\left(\frac{\partial\mathcal{W}}{\partial x}p_y - \frac{\partial\mathcal{W}}{\partial y}p_x\right)\right|Y_{\uparrow,\downarrow}\right\rangle > 0. \tag{14}$$

When $\Delta \neq 0$, it is convenient to use the basis

$$iS_\uparrow, iS_\downarrow, \frac{X_\uparrow + iY_\uparrow}{\sqrt{2}}, \frac{X_\downarrow - iY_\downarrow}{\sqrt{2}}, -\frac{X_\downarrow + iY_\downarrow - 2Z_\uparrow}{\sqrt{6}}, \frac{X_\uparrow - iY_\uparrow + 2Z_\downarrow}{\sqrt{6}}, -\frac{X_\uparrow - iY_\uparrow - Z_\downarrow}{\sqrt{3}}, \frac{X_\downarrow + iY_\downarrow + Z_\uparrow}{\sqrt{3}}, \tag{15}$$

which diagonalizes the valence bands if $\boldsymbol{B}_{\mathrm{eff}} = 0$.

Employing Eqs. (12), (14), we compute the EFA representation of $H$ of Eq. (9) in this basis. This gives us the Kane (K) Hamiltonian

$$H^{(\mathrm{K})} = \begin{pmatrix} H_s & h \\ h^\dagger & H_g + H_p \end{pmatrix}, \tag{16}$$

where $H_s = E_s + \hbar^2\boldsymbol{k}^2/2m + V + \mu_{\mathrm{B}}\boldsymbol{\sigma}\boldsymbol{B}_{\mathrm{eff}}$ and $H_p = E_s + \hbar^2\boldsymbol{k}^2/2m + V + \mu_{\mathrm{B}}\boldsymbol{\Sigma}\boldsymbol{B}_{\mathrm{eff}}$ act on the $s$ and $p$ functions respectively,

$$h = \frac{P}{\sqrt{3}}\begin{pmatrix} \sqrt{\frac{3}{2}}\widetilde{k}_+ & 0 & \sqrt{2}\widetilde{k}_z & \frac{1}{\sqrt{2}}\widetilde{k}_- & -\widetilde{k}_- & \widetilde{k}_z \\ 0 & \sqrt{\frac{3}{2}}\widetilde{k}_- & -\frac{1}{\sqrt{2}}\widetilde{k}_+ & \sqrt{2}\widetilde{k}_z & \widetilde{k}_z & \widetilde{k}_+ \end{pmatrix}, \tag{17}$$

with $\widetilde{k}_\pm = \widetilde{k}_x \pm i\widetilde{k}_y$, and

$$H_g = -\operatorname{diag}\left(E_g, E_g, E_g, E_g, E_g + \Delta, E_g + \Delta\right), \tag{18}$$

is a diagonal matrix that quantifies the gap between the bands. Here $E_g = E_s - E_p - \Delta/3$ and $\Sigma$ is a vector of the spin matrices for the $p$ functions. Expressions for its components can be found in Appendix A. Note also that $E_g = \xi - \Delta/3$.

## 3 Diagonalization of the Kane model

### 3.1 Unitary transformation: General formulation

We will now formulate a scheme that can be used to diagonalize (perturbatively) any Hamiltonian of the form of Eq. (16). For ordinary matrices, such a procedure is described in, e.g., Ref. [2]. We, on the other hand, are interested in diagonalizing operator valued matrices. Then it is preferable to modify the approach of Ref. [2]. We do this using the language of unitary transformations and operator identities. One of the advantages of this approach is that it allows to transform operators of all observables in a unified fashion.

Let us split the Hamiltonian as $H^{(\mathrm{K})} = H_{\mathrm{d}}^{(\mathrm{K})} + H_{\mathrm{od}}^{(\mathrm{K})}$, where

$$H_{\mathrm{d}}^{(\mathrm{K})} = \begin{pmatrix} H_s & 0 \\ 0 & H_g + H_p \end{pmatrix}, \qquad H_{\mathrm{od}}^{(\mathrm{K})} = \begin{pmatrix} 0 & h \\ h^\dagger & 0 \end{pmatrix}, \tag{19}$$

are its diagonal (d) and off-diagonal (od) components, respectively. We want to diagonalize $H^{(\mathrm{K})}$ by applying a unitary transformation. For this purpose, we define a unitary operator $U$ as

$$U = e^{iW}, \qquad W = \begin{pmatrix} 0 & w \\ w^\dagger & 0 \end{pmatrix}. \tag{20}$$

Diagonal and off-diagonal components of the unitary transformed Hamiltonian $H_U^{(\mathrm{K})} = U^\dagger H^{(\mathrm{K})} U$ are then represented by [2]

$$H_{U,\mathrm{d}}^{(\mathrm{K})} = H_{\mathrm{d}}^{(\mathrm{K})} + \sum_{k=1}^{\infty}\left[\frac{\left[H_{\mathrm{od}}^{(\mathrm{K})}, iW\right]^{(2k-1)}}{(2k-1)!} + \frac{\left[H_{\mathrm{d}}^{(\mathrm{K})}, iW\right]^{(2k)}}{(2k)!}\right], \tag{21a}$$

$$H_{U,\mathrm{od}}^{(\mathrm{K})} = H_{\mathrm{od}}^{(\mathrm{K})} + \sum_{k=1}^{\infty}\left[\frac{\left[H_{\mathrm{d}}^{(\mathrm{K})}, iW\right]^{(2k-1)}}{(2k-1)!} + \frac{\left[H_{\mathrm{od}}^{(\mathrm{K})}, iW\right]^{(2k)}}{(2k)!}\right], \tag{21b}$$

where the notation

$$[A, B]^{(n)} = [A, \underbrace{[B], B], \dots B}_{n \text{ times}}], \tag{22}$$

is used.

We need to solve the equation $H_{U,\mathrm{od}}^{(\mathrm{K})} = 0$ for $w$. The general perturbative solution can be found from Eq. (21b) as an expansion with respect to the inverse powers of $H_g$. We will consider only the terms up to the order $H_g^{-2}$. With this accuracy,

$$H_{U,\mathrm{od}}^{(\mathrm{K})} = 0 \iff H_{\mathrm{od}}^{(\mathrm{K})} + i\left[H_{\mathrm{d}}^{(\mathrm{K})}, W\right] = 0. \tag{23}$$

The last equation is equivalent to the equation $w = -ihH_g^{-1} + (H_s w - wH_p)H_g^{-1}$. We solve it

iteratively. The first iteration gives $w = -ihH_g^{-1}$, whereas the second one leads to

$$w = -ihH_g^{-1} + ihH_g^{-1}H_pH_g^{-1} - iH_shH_g^{-2}. \tag{24}$$

Next iterations produce contributions of the order $H_g^{-3}$ and higher. Hence, with $w$ of Eq. (24), the Hamiltonian $H_U^{(K)}$ can be assumed diagonal with the accuracy of up to the order $H_g^{-2}$. Consequently, we have determined the unitary transformation $U$ that we were looking for.

## 3.2 Unitary transformed Hamiltonian and observables: General formulation

We will denote the diagonal components of $H_U^{(K)}$ by $H^{(c)}$ and $H^{(v)}$, keeping in mind the upcoming application of this scheme to the description of the conduction (c) and the valence (v) bands of the Kane model. From Eq. (21a), we find

$$H_U^{(K)} = \begin{pmatrix} H^{(c)} & 0 \\ 0 & H^{(v)} \end{pmatrix} = \begin{pmatrix} H_s + i\left[hw^\dagger - wh^\dagger\right]/2 & 0 \\ 0 & H_g + H_p + i\left[h^\dagger w - w^\dagger h\right]/2 \end{pmatrix}. \tag{25}$$

For the conduction band, one can derive from Eqs. (24) and (25)

$$H^{(c)} = H_s - hH_g^{-1}h^\dagger + \frac{1}{2}\left(hH_g^{-1}H_pH_g^{-1}h^\dagger - H_shH_g^{-2}h^\dagger + h.c.\right), \tag{26}$$

which, up to the order $H_g^{-2}$, coincides with Eq. (8) of Ref. [20] formulated for time-independent $h$.

It is important that operators of other observables, like velocity and position, should also be transformed, on equal footing with the Hamiltonian. Otherwise important contributions to, for example, transport effects can be overlooked [24]. For an arbitrary observable $F$ and its unitary transformed version $F_U = U^\dagger F U$, we use the notations

$$F = \begin{pmatrix} F_s & f \\ f^\dagger & F_p \end{pmatrix}, \qquad F_U = \begin{pmatrix} F^{(c)} & \delta f \\ \delta f^\dagger & F^{(v)} \end{pmatrix}, \tag{27}$$

with the same subscripts and superscripts as for the Hamiltonian. Employing Eqs. (20) and (27), one can expand the "conduction band projection" of the observable $F$ as

$$F^{(c)} = F_s - \frac{1}{2}\left\{F_s, ww^\dagger\right\} + wF_pw^\dagger + i\left(fw^\dagger - wf^\dagger\right), \tag{28}$$

where $\{\cdot, \cdot\}$ denotes the anticommutator. Substitution of Eq. (24) into Eq. (28) followed by an expansion of the obtained expression up to $H_g^{-2}$ will give the desired result for $F^{(c)}$, generalizing Eq. (11) of Ref. [20] to the situation when $f \neq 0$.

In this work, we do not consider the remaining elements of $F_U$. However, for completeness, we present the expansions for $F^{(v)}$ and $\delta f$ in Appendix B. They can be used to analyze more complex models than the Kane model.

## 3.3 Hamiltonian of the conduction band (in the Kane model)

Let us derive the expression for the conduction band Hamiltonian $H^{(c)}$ in the Kane model by applying the general results of the previous sections to the Kane Hamiltonian $H^{(K)}$ given by Eq. (16). For this, we can use the well-known parametrization [14, 20]

$$h_\alpha H_g^{-n} h_\beta^\dagger = (-1)^n \left[\kappa_n \delta_{\alpha\beta} + i\lambda_n \sum_\gamma \epsilon_{\alpha\beta\gamma}\sigma_\gamma\right], \tag{29a}$$

$$\kappa_n = \frac{P^2}{3}\left[\frac{1}{(E_g + \Delta)^n} + \frac{2}{E_g^n}\right], \quad \lambda_n = \frac{P^2}{3}\left[\frac{1}{(E_g + \Delta)^n} - \frac{1}{E_g^n}\right], \tag{29b}$$

which can be also obtained directly, employing the results of Sec. 2.4. Here $\delta$ is the Kronecker delta, $\epsilon$ represents the Levi-Civita symbol, and $h_\alpha = \partial h / \partial \widetilde{k}_\alpha$. Note that $h$ depends on the components of $\widetilde{k}$ linearly (see Eq. (17)).

To compute $H^{(c)}$, the expressions of Eqs. (29) should be substituted into Eq. (26). The next steps mostly repeat those of Ref. [20]. The main difference with Ref. [20] is that the correct expression for $h$, as we have shown in Sec. 2.3, depends on $\widetilde{k}$ rather than on $k$ (see Eqs. (13), (17)). The difference between these two vectors is proportional to $\xi^{-1} = (E_g + \Delta/3)^{-1}$. We are interested in the accuracy of up to the order $E_g^{-2}$. Hence, assuming that $E_g$ and $\Delta$ are comparable, we see that the second term on the right-hand side of Eq. (26) should be computed using the full expressions for the components of $\widetilde{k}$. At the same time, for the terms in the parenthesis there, the deviation of $\widetilde{k}$ from $k$ can be neglected.

We also note that, for finite $B$ and $B_{\text{exc}}$, one more parameter is required to describe $H^{(c)}$,

$$\lambda = -\frac{2P^2}{9}\left[\frac{1}{E_g + \Delta} - \frac{1}{E_g}\right]^2, \tag{30}$$

in addition to those of Eq. (29b). It emerges from the Zeeman terms in $H_s$ and $H_p$ in the parenthesis of Eq. (26). Lengthy but straightforward computation allows us to obtain the following result,

$$\begin{aligned}
H^{(c)} = E_s &+ \frac{\hbar^2 k^2}{2m^*} + \lambda_{\text{SOC}}\left[\nabla V \times k\right]\sigma + V + \mu_B \sigma \left(\frac{g^*}{2}B + B_{\text{exc}}\right) \\
&+ \mu_B\left[\lambda_+ k^2 (\sigma B) + \frac{\lambda_-}{2}\left\{(kB),(k\sigma)\right\}\right] + \lambda \mu_B\left[k^2(\sigma B_{\text{exc}}) + \frac{1}{2}\left\{(kB_{\text{exc}}),(k\sigma)\right\}\right],
\end{aligned} \tag{31}$$

where we neglected terms quadratic with respect to $\mu_B$ and introduced the notations

$$\lambda_{\text{SOC}} = \lambda_2 + \frac{2\lambda_1}{\xi}, \quad \lambda_\pm = \lambda \pm 2\lambda_2 \pm \frac{2\lambda_1}{\xi}, \quad \frac{g^*}{2} = 1 + \frac{2m\lambda_1}{\hbar^2}, \quad \frac{1}{m^*} = \frac{1}{m} + \frac{2\kappa_1}{\hbar^2}. \tag{32}$$

Note that we do not consider any derivatives of $V$ and $A$ beyond the first in this work.

The last two expressions in Eq. (32) represent the effective $g$-factor and the effective mass respectively. They are well-known [14, 20, 29]. The renormalization of mass due to the exchange field $B_{\text{exc}}$, which is given by the second square bracketed group of terms in Eq. (31) is, as far as we know, derived here for the first time. The first square bracketed group in Eq. (31) represents the renormalization of mass by the magnetic field $B$ and is quantified by the parameters $\lambda_\pm$. The full expressions for them, to the best of our knowledge, were also missing so far. We note that the renormalization of mass by the magnetic field can be also interpreted as a $k$-dependent renormalization of the spin $g$-factor.

The spin-orbit coupling constant $\lambda_{\text{SOC}}$, in the Kane model, is commonly assumed to be equal to $\lambda_2$ [2, 14, 18, 20, 25, 27]. This result however misses the contributions from the interband matrix elements of the scalar potential $V$, as we have just demonstrated. Proper consideration of the latter leads to an additional contribution to $\lambda_{\text{SOC}}$ provided by $2\lambda_1/\xi$. Similarly, the interband matrix elements of the vector potential $A$ contribute to the renormalization of mass quantified by the parameters $\lambda_\pm$.

Let us estimate the magnitude of the computed correction to $\lambda_{\text{SOC}}$. For InSb at zero temperature, one can take $\Delta = 0.8$ eV and $E_g = 0.23$ eV [30]. This gives $(2\lambda_1/\xi)/\lambda_2 \approx 0.76$, which is a 76% change. For GaAs, again at zero temperature, the choice of $\Delta = 0.34$ eV and $E_g = 1.52$ eV [30] leads to $(2\lambda_1/\xi)/\lambda_2 \approx 1.02$. Hence, for GaAs the correct value of the SOC constant at $T = 0$ is approximately twice larger than it is commonly assumed to be. In general,

$(2\lambda_1/\xi)/\lambda_2$ varies from 0 to $18 - 12\sqrt{2} \approx 1.03$ (as a function of positive parameters $E_g$ and $\Delta$). Using data from Ref. [30], we tabulate this ratio for a set of materials in Table 1.

The obtained formula for $\lambda_{SOC}$ of Eq. (32) and the performed numerical estimates based on it provide the first main result of this paper.

Table 1: Zero temperature values of $(2\lambda_1/\xi)/\lambda_2$ for several materials.

|  | InSb | GaAs | InAs | InP |
|---|---|---|---|---|
| $(2\lambda_1/\xi)/\lambda_2$ | 0.76 | 1.02 | 1.01 | 1.01 |

## 3.4 Assumption of slow variation

On a technical level, the computed corrections to the SOC parameter and to the renormalization of mass are nonzero because we use two different "slow variation" assumptions for the envelope functions and for the scalar and vector potentials. Indeed, since the behaviour of the wave functions of Eq. (3) on the atomic length scale is encoded solely by the cell-periodic functions, one has to assume that the envelope functions are constant within each unit cell. As a result, they are pulled out of all the integrals corresponding to averaging over the cells.

At the same time, when one uses the same assumption for the potentials, their interband matrix elements are effectively disregarded and information about the related interband transitions is lost. This is why we use a different "slow variation" assumption for the potentials. Namely we allow them to vary smoothly within the cells. Their matrix elements are then computed as Taylor expansions. The interband matrix elements are proportional to the first spatial derivatives of the potentials. In this work, all higher derivatives are assumed small, therefore the expansions are well-defined.

# 4 Velocity and position in the Kane model

## 4.1 General relation between velocity and position in a crystal

Having derived and analyzed the conduction band Hamiltonian, we can now focus on the velocity operator, which is an essential object in studying transport phenomena. In the standard formulation, the velocity operator is a time-derivative of the position operator. In the Schrödinger picture, we have

$$\boldsymbol{v} = [\boldsymbol{r}, H]/i\hbar = (\boldsymbol{r}H - H\boldsymbol{r})/i\hbar. \tag{33}$$

Any effective description of a crystal takes into account only a finite number of bands. This means that instead of considering the entire Hilbert space (for single particle states), we have to project all states and all operators onto a certain subspace. This is achieved with the help of a projection operator which we can denote for example as $P$. Then, in this effective description, an arbitrary observable $F$ is approximated by $PFP$.

The velocity, position, and Hamiltonian are thus replaced by $P\boldsymbol{v}P$, $P\boldsymbol{r}P$, and $PHP$, respectively. However, the relation of Eq. (33) is not valid for these projected operators, because $P^2 \neq 1$ and we cannot insert $P^2$ between $\boldsymbol{r}$ and $H$ in

$$P\boldsymbol{v}P = P(\boldsymbol{r}H - H\boldsymbol{r})P/i\hbar \neq P(\boldsymbol{r}P^2 H - HP^2\boldsymbol{r})P/i\hbar = [P\boldsymbol{r}P, PHP]/i\hbar. \tag{34}$$

Therefore, the velocity operator in the "projected" theory should be defined as $P[\boldsymbol{r}, H]P/i\hbar$ rather than as a commutator of the projected coordinate and Hamiltonian, $[P\boldsymbol{r}P, PHP]/i\hbar$.

This simple idea was, in principle, formulated in 1973 by Noziéres and Lewiner in Ref. [20].[6] The authors explained that insertion of $P^2$ between a pair of operators leads to an incorrect exclusion of virtual processes associated with the remaining bands. At the same time, Ref. [20] ignored the interband matrix elements of the position operator.[7] In the absence of such elements, it is actually allowed to put $P^2$ between $r$ and $H$ in the definition of the velocity projection, and this is what Noziéres and Lewiner effectively did (to compute the velocity operator of the conduction band of the diagonalized Kane model). The issue, however, is that the underlying assumption of vanishing interband matrix elements of $r$ is just incorrect.

Still, for some applications, it is strongly beneficial to represent the velocity as a commutator of a certain operator and the Hamiltonian. For example, one needs such a representation to project the Kubo-Středa formula for dc conductivity [21,31] to a subband [24]. It turns out that, instead of using the physical position operator $r$ for this purpose, one can employ a fictitious position operator, which, upon commuting with the Hamiltonian, defines velocity. Below, we consider both these position operators and the velocity operator in the Kane model and derive the expressions for them in the conduction band of the diagonalized version of the latter. These results can be used to formulate effective transport theories in narrow-gap semiconductors.

### 4.2 Velocity and fictitious position operator

To analyze velocity in the Kane model, we first need to compute the velocity operator of the original crystal Hamiltonian. This is carried out by differentiating either Eq. (8) or Eq. (9) with respect to $p$ and multiplying the SOC contribution to the result by the factor of 2 [24],[8]

$$v = \frac{p}{m} + \frac{\hbar k}{m} - \frac{e}{2mc}[B \times \rho] - \frac{\hbar}{2m^2c^2}[\nabla \mathcal{W} \times \sigma] . \tag{35}$$

The last term here is a vacuum SOC correction to the first term and can be neglected. Once this is done, the velocity operator becomes $v = \hbar^{-1}\partial H/\partial k$. To obtain its 8-band EFA representation, which we denote by $v^{(K)}$, one should compute the elements $\langle U_i|v|U_j\rangle$. Integration over unit cells does not interfere with differentiating over $k$, and we thus infer that

$$v^{(K)} = \frac{1}{\hbar}\frac{\partial H^{(K)}}{\partial k} . \tag{36}$$

We would like to express the right-hand side of the above as a commutator, in analogy with the standard recipe used to define velocity. This can be achieved with the help of the fictitious position operator $\mathfrak{r}$ which is dual to the operator $k$ in a sense that $[k_\alpha, \mathfrak{r}_\beta] = -i\delta_{\alpha\beta}$. In the 8-band basis, we can understand $\mathfrak{r}$ as a diagonal matrix with all elements being equal to the physical coordinate $r$. The latter is also the argument of the envelope functions $\psi_i(r)$. We will denote this operator by $\mathfrak{r}^{(K)}$. Clearly, $\mathfrak{r}^{(K)}$ is just a physical position operator with the nullified interband matrix elements (see also Eq. (38) below). Using the commutation property, one finds

$$v^{(K)} = [\mathfrak{r}^{(K)}, H^{(K)}]/i\hbar . \tag{37}$$

We will use this relation to compute the conduction band velocity in the next Section.

Before we do this, let us highlight the difference between $\mathfrak{r}^{(K)}$ and the physical position operator $r^{(K)}$. The two operators share the same diagonal elements, however the latter has

---

[6]The formulation can be found between Eq. (11) and Eq. (12) of that work.

[7]In fact, together with the interband matrix elements of the scalar and vector potentials.

[8]Note that we ignore other vacuum relativistic effects. The only remaining signature of such is the SOC-induced splitting of the valence bands. Its effect on velocity however is negligible.

also the interband ones. Employing the results of Sections 2.3, 2.4, we can write

$$\mathfrak{r}^{(\mathrm{K})} = \begin{pmatrix} \boldsymbol{r} & 0 \\ 0 & \boldsymbol{r} \end{pmatrix}, \qquad \boldsymbol{r}^{(\mathrm{K})} = \begin{pmatrix} \boldsymbol{r} & -i(\xi)^{-1}\partial h/\partial\widetilde{\boldsymbol{k}} \\ i(\xi)^{-1}\partial h^{\dagger}/\partial\widetilde{\boldsymbol{k}} & \boldsymbol{r} \end{pmatrix}, \tag{38}$$

where $h$ is defined by Eq. (17).

### 4.3 Velocity operator and position operators in the conduction band

For the components of the unitary transformed velocity $\boldsymbol{v}_U^{(\mathrm{K})} = U^{\dagger}\boldsymbol{v}^{(\mathrm{K})}U$ and fictitious position $\mathfrak{r}_U^{(\mathrm{K})} = U^{\dagger}\mathfrak{r}^{(\mathrm{K})}U$, we use the notation

$$\boldsymbol{v}_U^{(\mathrm{K})} = \begin{pmatrix} \boldsymbol{v}^{(\mathrm{c})} & \delta\boldsymbol{v} \\ \delta\boldsymbol{v}^{\dagger} & \boldsymbol{v}^{(\mathrm{v})} \end{pmatrix}, \qquad \mathfrak{r}_U^{(\mathrm{K})} = \begin{pmatrix} \mathfrak{r}^{(\mathrm{c})} & \delta\mathfrak{r} \\ \delta\mathfrak{r}^{\dagger} & \mathfrak{r}^{(\mathrm{v})} \end{pmatrix}. \tag{39}$$

We are interested in computing $\boldsymbol{v}^{(\mathrm{c})}$. Of course, this can be done directly, by substituting the explicit expressions for the components of the velocity operator $\boldsymbol{v}^{(\mathrm{K})}$ into Eq. (28). However, an easier way is to make use of Eq. (37). Indeed, unitary transformations preserve commutators, and $H_U^{(\mathrm{K})}$ is diagonal. Therefore from $\boldsymbol{v}_U^{(\mathrm{K})} = [\mathfrak{r}_U^{(\mathrm{K})}, H_U^{(\mathrm{K})}]/i\hbar$ one can derive a simple relation that determines $\boldsymbol{v}^{(\mathrm{c})}$,

$$\boldsymbol{v}^{(\mathrm{c})} = [\mathfrak{r}^{(\mathrm{c})}, H^{(\mathrm{c})}]/i\hbar. \tag{40}$$

For $\mathfrak{r}^{(\mathrm{c})}$, Eq. (28) takes a particularly concise form because $\mathfrak{r}^{(\mathrm{K})}$ is diagonal. Employing Eq. (24), we find to the order $E_{\mathrm{g}}^{-2}$:

$$\mathfrak{r}^{(\mathrm{c})} = \boldsymbol{r} + \frac{1}{2}\left([h,\boldsymbol{r}]H_{\mathrm{g}}^{-2}h^{\dagger} + h.c.\right) = \boldsymbol{r} + \lambda_2[\boldsymbol{k}\times\boldsymbol{\sigma}], \tag{41}$$

where we made use of Eqs. (17) and (29). This result is Eq. (22) of Ref. [20]. In that work, however, $\mathfrak{r}^{(\mathrm{c})}$ was understood as a physical position operator in the conduction band, which is not correct. The correct expression for the physical position operator can be computed similarly to the fictitious one,

$$\boldsymbol{r}^{(\mathrm{c})} = \boldsymbol{r} + \frac{1}{2}\left([h,\boldsymbol{r}]H_{\mathrm{g}}^{-2}h^{\dagger} + h.c.\right) + \frac{i}{\xi}\left(\frac{\partial h}{\partial\widetilde{\boldsymbol{k}}}H_{\mathrm{g}}^{-1}h^{\dagger} - h.c.\right) = \boldsymbol{r} + \lambda_{\mathrm{SOC}}[\boldsymbol{k}\times\boldsymbol{\sigma}], \tag{42}$$

where we used Eqs. (24), (28), (29), (38). Thus, in the conduction band, the two position operators differ by the expressions for the SOC strength. One of them is quantified by the constant $\lambda_2$, while another depends on $\lambda_{\mathrm{SOC}}$. Note that the difference between the two comes solely from the interband matrix elements of $\boldsymbol{r}$.

By commuting the right-hand side of Eq. (41) with $H^{(\mathrm{c})}$ given by Eq. (31), we arrive at

$$\boldsymbol{v}^{(\mathrm{c})} = \frac{\hbar\boldsymbol{k}}{m^*} - \frac{1}{\hbar}(\lambda_{\mathrm{SOC}} + \lambda_2)[\boldsymbol{\nabla}V\times\boldsymbol{\sigma}], \tag{43}$$

where $\boldsymbol{B} = \boldsymbol{B}_{\mathrm{exc}} = 0$ is assumed for simplicity. The full expression for the velocity operator $\boldsymbol{v}^{(\mathrm{c})}$ in the presence of the magnetic and exchange fields can be found in Appendix C.

The term $[\boldsymbol{\nabla}V\times\boldsymbol{\sigma}]$ in the expression for $\boldsymbol{v}^{(\mathrm{c})}$ is a SOC correction. It is instructive to trace the origin of the contributions to the prefactor in front of it. The first one, proportional to $\lambda_{\mathrm{SOC}}$, comes from a commutator of $\boldsymbol{r}$ in Eq. (41) and the third term on the right-hand side of Eq. (31). The second contribution originates from the commutator of $\lambda_2[\boldsymbol{k}\times\boldsymbol{\sigma}]$ in Eq. (41) and $V$ in Eq. (31). From Eqs. (31), (43), we see that the ratio between the SOC constant of the velocity operator and the SOC constant of the Hamiltonian equals $1 + \lambda_2/\lambda_{\mathrm{SOC}}$. In a

system described by the Pauli Hamiltonian, $\lambda_{\text{SOC}} = \lambda_2$[9] and thus this ratio is 2 (which is a known result [24, 32–34]). In the Kane model, as it turns out, the situation is different. As a function of the parameters $E_g$ and $\Delta$, the expression $1 + \lambda_2/\lambda_{\text{SOC}}$ can take any value between $4(23 + 3\sqrt{2})/73 \approx 1.49$ and 2. For GaAs at zero temperature, it is very close to the lower bound. In InSb, it is approximately 1.57.

In Table 2, we list the expressions for the SOC strengths in the Kane model and in the Pauli system. Together with Eqs. (41), (42), (43), they constitute the second most important result of this paper.

Table 2: SOC constants for a system described by the Pauli Hamiltonian, the Kane model, InSb, and GaAs. In the three latter cases, we consider operators in the conduction band of the diagonalized Kane model. For InSb and GaAs, zero temperature is assumed. We introduced the parameter $\lambda_{\text{vac}} = \hbar^2/4m^2c^2$, whereas $\lambda_{\text{SOC}} = \lambda_2 + 2\lambda_1/\xi$ and $\xi = E_g + \Delta/3$ were already defined in the text. For the definition of $\lambda_i$ see Eq. (32).

| | Hamiltonian $\dfrac{\hbar^2 k^2}{2m^*} + \eta_1[\boldsymbol{\nabla} V \times \boldsymbol{k}]\boldsymbol{\sigma}$ | Velocity $\dfrac{\hbar \boldsymbol{k}}{m^*} - \dfrac{\eta_2}{\hbar}[\boldsymbol{\nabla} V \times \boldsymbol{\sigma}]$ | Fictitious position $\boldsymbol{r} + \eta_3[\boldsymbol{k} \times \boldsymbol{\sigma}]$ | Physical position $\boldsymbol{r} + \eta_4[\boldsymbol{k} \times \boldsymbol{\sigma}]$ |
|---|---|---|---|---|
| Pauli system | $\eta_1 = \lambda_{\text{vac}}$ | $\eta_2 = 2\lambda_{\text{vac}}$ | $\eta_3 = \lambda_{\text{vac}}$ | $\eta_4 = \lambda_{\text{vac}}$ |
| Kane model | $\eta_1 = \lambda_{\text{SOC}}$ | $\eta_2 = \lambda_{\text{SOC}} + \lambda_2$ | $\eta_3 = \lambda_2$ | $\eta_4 = \lambda_{\text{SOC}}$ |
| InSb, $T = 0$ | $\eta_1 = \lambda_{\text{SOC}}$ | $\eta_2 \approx 1.57\,\lambda_{\text{SOC}}$ | $\eta_3 \approx 0.57\,\lambda_{\text{SOC}}$ | $\eta_4 = \lambda_{\text{SOC}}$ |
| GaAs, $T = 0$ | $\eta_1 = \lambda_{\text{SOC}}$ | $\eta_2 \approx 1.49\,\lambda_{\text{SOC}}$ | $\eta_3 \approx 0.49\,\lambda_{\text{SOC}}$ | $\eta_4 = \lambda_{\text{SOC}}$ |

## 4.4 Kubo formula for dc conductivity

Recently we revised the Kubo formula for dc conductivity in systems with SOC. In particular we found a novel contribution to the conductivity tensor that we suggested to denote by $\sigma_{\alpha\beta}^{\text{III}}$ in Eq. (24c) of Ref. [24]. There, this contribution was proportional to $[r_\alpha, r_\beta]$, where $\boldsymbol{r}$ referred to the operator that, upon commutation with the Hamiltonian, defined the velocity operator. In vacuum, the role of $\boldsymbol{r}$ is played by the physical position operator. In the Kane model, as we have demonstrated, $\boldsymbol{r}$ should be replaced by $\mathfrak{r}$. Therefore, in this case, $\sigma_{\alpha\beta}^{\text{III}}$ is determined by $[\mathfrak{r}_\alpha, \mathfrak{r}_\beta]$. It is important not to confuse the fictitious position operator here with the physical one.

## 4.5 SOC constants in the Rashba model

Quantum well heterostructures with the structural inversion asymmetry are often described by the two-dimensional (2D) Rashba model [23]. In such systems, electrons are confined by an asymmetric scalar potential and experience the Rashba-type SOC splitting of the bands. In general, the strength of this splitting is determined by both the well material and the barrier penetration [35, 36]. In certain cases, however, the penetration is weak[10] and the infinite barrier approximation [37] can be used to compute the SOC splitting. In this approach, only the well material is analyzed while the barriers are modeled as perfect insulators [35].

Let us assume that the heterostructure well material is described by the Kane model. As our results suggest, the 2D Rashba (R) model derived for this heterostructure under the infinite barrier approximation will be such that the Hamiltonian and the velocity operator are

---

[9]We mean that in such systems the SOC constant that enters the expression for the position operator coincides with the one that quantifies the SOC term in the Hamiltonian. Note that the physical position operator and the fictitious position operator do not differ for such systems. See also Table 2.

[10]For example, the barrier penetration can be reduced by gating [36].

written as

$$H^{(\mathrm{R})} = \frac{\hbar^2 \boldsymbol{k}^2}{2m^*} + \alpha_{\mathrm{R}} \hbar \left[ \boldsymbol{k} \times \boldsymbol{\sigma} \right]_z \,, \qquad \boldsymbol{v}^{(\mathrm{R})} = \frac{\hbar \boldsymbol{k}}{m^*} + \gamma \alpha_{\mathrm{R}} \left[ \boldsymbol{\sigma} \times \boldsymbol{e}_z \right] \,, \tag{44}$$

where $\gamma$ ranges from $4(23 + 3\sqrt{2})/73 \approx 1.49$ to 2 and $\alpha_{\mathrm{R}}$ is proportional to the average gradient of the confining potential. We emphasize that $\gamma \neq 1$ in Eq. (44). This important fact, for example, prevents the famous cancellation of the anomalous Hall effect in the metallic regime of the 2D Rashba model [24]. Other transport phenomena originating in the Rashba coupling should be affected by this too.

Computation of the barrier penetration contributions to $\boldsymbol{v}^{(\mathrm{R})}$ is a complex problem that requires separate investigation. However, as a basis postulate, we suggest to assume that there is no general relation between the SOC constants of the Hamiltonian and the velocity operator.

Eq. (44) delivers our third main result.

## 4.6 Connection with experiments

It seems that the most realistic way of probing our results experimentally would be by measuring the Rashba SOC strength in heterostructures. To date, multiple works have reported significant discrepancies between the theoretically predicted values of the Rashba coupling and the experimentally measured ones. The latter are often 2-3 times larger than the former [38–41]. For GaAs, InAs, and InP, our theory predicts a twofold increase of the spin-orbit coupling parameter $\lambda_{\mathrm{SOC}}$ in the Hamiltonian (and a slightly smaller increase for InSb). It seems plausible that this may resolve the disagreement between the theory and experiment. To verify our prediction, one can use the results of Ref. [35], properly generalized to allow for finite interband matrix elements of the scalar potential.

Another effect that might be measurable experimentally is the renormalization of the spin $g$-factor by the first square bracketed group in Eq. (31). One can expect to observe the density dependence of the $g$-factor, not related to electron-electron interactions [42, 43] or the orbital magnetism [44].

## 5 Conclusion

Using unitary transformations and operator identities, we formulated a scheme suitable for a perturbative diagonalization of Hamiltonians with a block structure. We used it to analyze the 8-band Kane model in the presence of the interband matrix elements of the scalar potential, vector potential, and position operator. We obtained expressions for the SOC constants that quantify corrections to the operators in the conduction band of the diagonalized Kane model. These expressions differ from those that can be found in the literature. For GaAs at zero temperature, the SOC term in the Hamiltonian is parameterized by a constant that is approximately twice larger than it is commonly assumed. We also explained why the velocity operator in a crystal should not be computed as a commutator of the physical position operator and the Hamiltonian. However, often one can just replace the physical position operator with a fictitious one in the corresponding definition. We also derived the expressions for the magnetic field induced renormalization of mass. The theory developed here forms the basis for study of transport phenomena in spin-orbit coupled systems, in particular using the Rashba model.

## Acknowledgments

We are grateful to Sasha Rudenko and Misha Katsnelson for informative discussions. We thank Bertrand Halperin and Dimitrie Culcer for thorough reviews that helped us improve the manuscript.

**Funding information** I. A. A. and R. A. D. have received funding from the European Research Council (ERC) under the European Union's Horizon 2020 research and innovation programme (Grant No. 725509). The Research Council of Norway (RCN) supported A. B. through its Centres of Excellence funding scheme, project number 262633, "QuSpin", and RCN project number 323766. M. T. has received funding from the European Union's Horizon 2020 research and innovation program under the Marie Skłodowska-Curie grant agreement No 873028.

## A  Spin matrices for the valence bands

Spin matrices for the $p$ functions, in the basis of Eq. (15), are

$$
\Sigma_x = \begin{pmatrix}
0 & 0 & -\frac{1}{\sqrt{3}} & 0 & 0 & \sqrt{\frac{2}{3}} \\
0 & 0 & 0 & \frac{1}{\sqrt{3}} & -\sqrt{\frac{2}{3}} & 0 \\
-\frac{1}{\sqrt{3}} & 0 & 0 & \frac{2}{3} & \frac{\sqrt{2}}{3} & 0 \\
0 & \frac{1}{\sqrt{3}} & \frac{2}{3} & 0 & 0 & \frac{\sqrt{2}}{3} \\
0 & -\sqrt{\frac{2}{3}} & \frac{\sqrt{2}}{3} & 0 & 0 & \frac{1}{3} \\
\sqrt{\frac{2}{3}} & 0 & 0 & \frac{\sqrt{2}}{3} & \frac{1}{3} & 0
\end{pmatrix}, \quad
\Sigma_y = i \begin{pmatrix}
0 & 0 & \frac{1}{\sqrt{3}} & 0 & 0 & -\sqrt{\frac{2}{3}} \\
0 & 0 & 0 & \frac{1}{\sqrt{3}} & -\sqrt{\frac{2}{3}} & 0 \\
-\frac{1}{\sqrt{3}} & 0 & 0 & -\frac{2}{3} & -\frac{\sqrt{2}}{3} & 0 \\
0 & -\frac{1}{\sqrt{3}} & \frac{2}{3} & 0 & 0 & \frac{\sqrt{2}}{3} \\
0 & \sqrt{\frac{2}{3}} & \frac{\sqrt{2}}{3} & 0 & 0 & \frac{1}{3} \\
\sqrt{\frac{2}{3}} & 0 & 0 & -\frac{\sqrt{2}}{3} & -\frac{1}{3} & 0
\end{pmatrix},
$$

and $\Sigma_z$ can be computed as $\Sigma_z = -i\Sigma_x \Sigma_y$.

## B  Components of observables in the diagonalized Kane model

The remaining components of the matrix on the right-hand side of Eq. (27) are determined by

$$
F^{(\mathrm{v})} = F_p - \frac{1}{2}\left\{F_p, w^\dagger w\right\} + w^\dagger F_s w + i\left(f^\dagger w - w^\dagger f\right), \tag{B.1a}
$$

$$
\delta f = f - \frac{1}{2}(fw^\dagger w + ww^\dagger f) + wf^\dagger w + i\left(F_s w - wF_p\right), \tag{B.1b}
$$

where $w$ is to be expanded in accordance with Eq. (24).

## C  Full expression for velocity

The full expression for the velocity operator in the conduction band of the diagonalized Kane model reads

$$
\begin{aligned}
\boldsymbol{v}^{(\mathrm{c})} = \frac{\hbar\boldsymbol{k}}{m^*} - \frac{1}{\hbar}(\lambda_{\mathrm{SOC}} + \lambda_2)\left[\boldsymbol{\nabla}V \times \boldsymbol{\sigma}\right] + \frac{\mu_{\mathrm{B}}}{\hbar}\Big[ & 2(\lambda_+ + \lambda_2)\boldsymbol{k}\,(\boldsymbol{\sigma}\boldsymbol{B}) + \lambda_-\boldsymbol{B}\,(\boldsymbol{k}\boldsymbol{\sigma}) \\
& + (\lambda_- - 2\lambda_2)\boldsymbol{\sigma}\,(\boldsymbol{k}\boldsymbol{B}) + 2\lambda\boldsymbol{k}\,(\boldsymbol{\sigma}\boldsymbol{B}_{\mathrm{exc}}) \\
& + (\lambda + 2\lambda_2)\boldsymbol{B}_{\mathrm{exc}}\,(\boldsymbol{k}\boldsymbol{\sigma}) + (\lambda - 2\lambda_2)\boldsymbol{\sigma}\,(\boldsymbol{k}\boldsymbol{B}_{\mathrm{exc}})\Big].
\end{aligned} \tag{C.1}
$$



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
