# Peer review of "Position operators and interband matrix elements of scalar and vector potentials in the 8-band Kane model"

_SciPost Physics, doi:SciPost Phys. 17, 009 (2024)_

## Round 2 · Referee Report · Bertrand Halperin (Referee 1) · 2024-2-18

Strengths

The paper revises long-standing theoretical understanding of effects of spin-orbit coupling on electrons in an important class of materials. The work is carefully done and has major conceptual significance.

Weaknesses

The connection with possible experiments is not clear.

Report

This is a very interesting paper that clearly deserves publication. The authors present compelling arguments that fundamental theoretical analyses of spin-orbit effects in small-band-gap III-V semiconductors, which have been widely accepted for many years, are incorrect because they have omitted important terms. Using the eight-band Kane model, the authors obtain corrected formulas for such quantities as the effective Hamiltonian for electrons in the conduction band in the presence of magnetic fields and/or an external scalar potential that varies slowly in space, spin-dependent anomalous terms in the carrier velocity, and subtle differences between two possible definitions of the position of an electron.

The formulas at issue are fundamental from a conceptual point of view, and they form a logical starting point for further investigations into experimental situations, where such assumptions as that of a potential that varies slowly in space may not be valid. It is not clear, however, whether there are consequences of the authors’ modifications that can be directly tested by experiments, either existing ones or ones that could be envisioned in the near future. Are the authors aware of any discrepancies between experiment and theory in the literature that might be resolved by their new analysis? It would be good if the authors could comment on these points.

In relation to possible comparisons between theory and experiment, I believe that Section 4.5 on SOC constants in the Rashba model requires some qualification. The Rashba coupling constant $\alpha_R$ for a two-dimensional electron gas in an asymmetric well cannot be calculated using the type of analysis proposed in this work. The effective confining potential $V{z}$ seen by an electron an actual well cannot be treated as slowly varying in space, as it varies on an atomic length scale at the interface between the well and the confining barrier. Moreover, if one were to use the Kane model Eq. 30 and to treat the potential as slowly varying in space, one would find that $\alpha_R=0$, because the expectation value $<dV/dz>$ is the average force on the wavefunction, which must vanish in equilibrium. A correct value of the Rashba coupling can only be obtained if one properly considers contributions from the portion of the electron wave function which enters the barrier layer, where the effective electric field will be large and the parameters such as the effective mass and the spin-orbit coupling will be different than in GaAs. Presumably, the effective operator for the electron position will also be affected by properties of the barrier material. The authors may be correct that the coefficients of the spin orbit terms in the Hamiltonian and the velocity in an asymmetric well are not related to each other by a universal constant, but it is misleading to use the Kane model as an argument for that.

The manuscript could benefit from clarifications at a few other points:

(a). When presenting Eq. (2), it would be helpful to explain that the wave functions $U_i$ are the solutions for states with zero wave vector, i.e., at the center of the Brillouin zone.

(b). I do not understand the meaning of Eq. (4). If $F$ is an arbitrary operator, and $\Psi$ is a function of the form (3), then the left-hand side of Eq. (4) is a function which depends strongly or position within a unit cell. The expression between the equal signs, however, is a constant, within any given unit cell. I do not know how to interpret the expression on the right-hand side because I do not understand the meaning of the symbol $<U_i|F|U_j>(r)$ , which appears there. The authors should remedy these problems. It seems likely, also, that the authors may have to put some restrictions on the operators $F$ they are considering, such as local operation and, perhaps, conditions on smoothness.

(c). At the beginning of Sec.2.2, the authors introduce the symbol $\rho$ to represent a vector within a single unit cell, describing the difference between some point in the cell and a reference point within the cell, such as the center of the cell. However, in Eq. (4), the same symbol was used represent an absolute position, presumably defined relative to a some fixed origin that could be far away from the point in question. I found this confusing, and I recommend that the authors use a different symbol, such as $r^\prime$ for the variable of integration in Eq. (4).

Requested changes

1). I request that the authors comment on possible connections to experiments, as suggested in my report.

(2). I suggest that the authors clarify there discussion of the asymmetrically-confined two-dimensional electron gas, as requested in my report

(3). I request that the authors address the points (a), (b), and (c), raised at the end of my report.

  • validity: -
  • significance: -
  • originality: -
  • clarity: -
  • formatting: -
  • grammar: -

Author:  Ivan Ado  on 2024-03-15  [id 4369]

(in reply to Report 1 by Bertrand Halperin on 2024-02-18)

We thank Prof. Halperin for a very thorough review and useful suggestions. Below, we comment on the issues raised in the report.

  1. First, let us clarify better the conditions we impose on the scalar and vector potentials. The standard approach assumes that they vary so slowly on the atomic length scale that both can be considered constant within each unit cell. Under such an assumption, the potentials are pulled out of all the integrals corresponding to averaging over the unit cells. Effectively, this means that only intraband matrix elements of the potentials are taken into account. We argue, however, that this assumption is excessive, in particular, because it automatically disregards the interband matrix elements of the potentials. If one eases the assumption by allowing the scalar potential V (or the vector potential A) to vary within the cells in a smooth way, its matrix elements can be computed as a Taylor expansion. The “leading order” contribution to the interband matrix elements will be proportional to dV/dr=V’. If |V’(r_1)-V’(r_2)| is much smaller than |V’(r)| for all r, r_1, r_2 that lie in the cell, the first term of the expansion is well defined (this is the case, e.g., for a linear function). If all higher derivatives of V can be considered small, the remaining terms of the expansion are neglected. This can be understood as a modified assumption of slow variation. We will supplement the text with an additional clarification of this subject.

  2. We agree with Prof. Halperin that Section 4.5 describing Rashba coupling must be formulated more accurately. The current formulation is valid for interfaces that can be described by an infinite barrier model. It might also be used when the barrier penetration is significantly reduced by gating. However, in general, contributions to the Rashba coupling from the barrier materials are nonzero and the Kane model alone is indeed insufficient to compute them.

  3. We note that the contribution to the conduction band Hamiltonian proportional to <dV/dz> does not have to vanish in equilibrium. This term is present only in the "unitary transformed theory" and originates from the electric field in the valence bands. The Ehrenfest theorem does not apply in this case.

  4. Following the suggestion by Prof. Halperin, we performed a literature study and found a number of works reporting discrepancies between the theoretically predicted values of the Rashba coupling and the experimentally measured ones. In many cases, the latter are 2-3 times larger than the former. For GaAs and InAs, our theory predicts a twofold increase of the spin-orbit coupling parameter \lambda in the Hamiltonian (and a slightly smaller increase for InSb). It seems plausible that this may resolve the disagreement between the theory and experiment. We will discuss this matter in the new version of the text.

  5. Regarding comment (a), we can add an extra specification that the Schrödinger equation (2) determines the cell-periodic functions. We, however, would prefer not to introduce any wave vectors at this point in the text to avoid possible confusion.

  6. In the next version of the paper, we will make the passage around Eqs. (3) and (4) less sketchy. The purpose of this passage is to describe how a projection to a band is performed in the Kane model. It is important that a single algorithm exists to perform such a projection for any operator and any state.

  7. We agree that replacing \rho with r’ in Eq.(4) can help in avoiding potential misinterpretation.

---

## Round 2 · Referee Report · Dimitrie Culcer (Referee 2) · 2024-4-15

Strengths

  1. Thorough calculation.
  2. It treats an issue which has generally been swept under the carpet in the community and the results may have far-reaching consequences.

Weaknesses

  1. Relation to observables not emphasised enough.
  2. The presentation is relatively formal, which is good for specialist theorists but not ideal for a general audience.

Report

The manuscript Position operators and interband matrix elements of scalar and vector potentials in the 8-band Kane model by Ado et al investigates the multi-band matrix elements of the position operator in the Kane model, which also affects the interaction with external electrostatic potentials and the orbital interaction with a magnetic field. The paper follows many of the principles of the Foldy-Wouthuysen transformation for Dirac electrons, but with several important differences brought about by the unit-cell structure inherent in crystal lattices. Whereas some of the starting ideas have been discussed in the community in a qualitative fashion previously, I am not aware of any systematic study and I believe the manuscript represents a strong contribution to the field.

The main novelty of this paper, as I understand it, is to stress the importance of inter-band matrix elements of the position operator in the context of the external potential \nabla V and the Peierls substitution. The latter have generally been treated in the literature in a hand-waving way without any justification. The authors are correct that neglecting such inter-band position terms in the interaction with electric and magnetic fields is inappropriate – I am aware of at least one quantitative example where this is the case. Precisely such an approximation is used in the well-known textbook by R. Winkler to conclude that T_d-symmetry terms are negligible, being 4-6 orders of magnitude smaller than conventional Rashba and Dresselhaus terms. Nevertheless, Philippopoulos et al [Phys. Rev. B 102, 075310 (2020)] recently showed that, when inter-orbital matrix elements are taken into account, the result is of the same order of magnitude as other inversion-breaking contributions. In fact there is no justification for neglecting inter-orbital/inter-band contributions to the position operator and the only compelling explanation is that until now this has simply been a matter of convenience.

In this work the authors focus on the conduction band. Unsurprisingly, when the full calculation is performed, the Rashba interaction in the conduction band is no longer parameterised by a single constant, but exhibits intra-band and inter-band contributions. Similarly, when this procedure is applied to the full Kane Hamiltonian with an external potential and the Peierls substitution, additional interaction terms emerge in the Hamiltonian projected onto the conduction band. These are given, for example by several terms in Eq. 30.

The authors also define an alternative position operator as the conjugate to the wave vector in the envelope function approximation for a finite Hilbert space, and the velocity operator resulting from it. This is essentially the side-jump term in the position operator, except it differs from the traditional side-jump term of Nozieres and Lewiner in the spin-orbit matrix element in the prefactor. The authors point out that this fictitious position operator in general does not coincide with the physical position operator.

There are some minor points:
- The authors state that inter-band matrix elements of \nabla V are neglected in all treatments they are aware of – it will be helpful to give a series of references for concreteness.
- It will help to restore dot products in the conventional way e.g. \sigma \cdot B, which will increase readability.
- Similarly it will help to tabulate the numerical contributions e.g. at the end of Section III \lambda_1 and \lambda_2 for different materials, in the same way it is done for the position operator in the current Table I.

I have a number of physics questions, and these are also related to the issue of readability highlighted above. Right now the paper has a fairly abstract tone and is likely to appeal mostly to theorists, whereas I believe the addition of some concrete predictions for measurable quantities will vastly increase its appeal to the experimental side of the spectrum. The Kane model is the workhorse of much of semiconductor physics, covering the conduction and valence bands of many semiconductors as well as topological materials, thus many conclusions drawn from it affect a substantial fraction of condensed matter physics. This fact could be exploited to write a short section on Experimental Observables, which may or may not be related to the following questions:
- I am somewhat surprised that the bracketed terms in Eq. 30 are referred to as renormalisations of the mass due to the magnetic field. To me they look like a correction to the Zeeman term where the g-factor has k^2 renormalisations. I would also stress the novelty of this term because such k^2 Zeeman terms do appear in hole systems (albeit with a different symmetry) but are not known in electron systems. Emphasising this concrete point would help broadcast the message to a wider readership – I imagine that many experimentalists would be interested in detecting a density-dependent g-factor in an electron system, where the density dependence would not be due to electron-electron interactions (small r_s).
- Following on from the idea above, what consequences would such a k^2 Zeeman term have for devising e.g. a Stoner criterion for the development of spontaneous ferromagnetism in systems described by the Kane model? Of course the Stoner criterion itself has a limited applicability, but conclusions drawn from it can be instructive.
- Similarly, what qualitative consequences are expected for the longitudinal and Hall magneto-resistances?
- Is there any system where the inter-band contributions can actually dominate?

In summary, the paper is timely and enlightening and should be considered for publication once the questions above are addressed.

Requested changes

  1. The authors state that inter-band matrix elements of \nabla V are neglected in all treatments they are aware of – it will be helpful to give a series of references for concreteness.
  2. It will help to restore dot products in the conventional way e.g. \sigma \cdot B, which will increase readability.
  3. Similarly it will help to tabulate the numerical contributions e.g. at the end of Section III \lambda_1 and \lambda_2 for different materials, in the same way it is done for the position operator in the current Table I.

Recommendation

Ask for minor revision

  • validity: top
  • significance: top
  • originality: top
  • clarity: top
  • formatting: perfect
  • grammar: perfect

Author:  Ivan Ado  on 2024-05-13  [id 4485]

(in reply to Report 2 by Dimitrie Culcer on 2024-04-15)

We thank Prof. Culcer for an extensive review, multiple suggestions, and also for bringing Phys. Rev. B 102, 075310 (2020) to our attention.

  1. In the new version of the manuscript, we will refer to a number of classic works that neglect the interband matrix elements of the position-dependent operators. We will also cite Phys. Rev. B 102, 075310 (2020) and rephrase the passage on the conventional treatment of the interband matrix elements of the scalar potential.

  2. We agree with Prof. Culcer that the paper will benefit if predictions for measurable quantities are provided. We also agree that the renormalization of mass by the magnetic field can be interpreted as a k-dependent renormalization of the spin g-factor. This may indeed provide a density dependence of the latter in experiments, not related to electron-electron interactions or the orbital magnetism. We will mention this in the new version of the text. Furthermore, we will comment on the discrepancies between the theoretically predicted values of the Rashba spin-orbit coupling and the measured ones. It might happen that our results help resolve the issue.

  3. We will tabulate the ratio (2λ1/ξ)/λ2 for a set of materials. This should also give a more intuitive understanding of the relative strength of the interband matrix elements. In the present model they do not dominate over the intraband once.

  4. Indeed, it should be interesting to consider magnetoresistance effects and Stoner instabilities in relation to the developed theory, but these topics lie significantly beyond the scope of the present work.

  5. We would prefer to avoid using dots for the scalar product operation as this will make some of the formulas too bulky. In particular, we will have to sacrifice the single line formatting of Eq. (8) if dots are used.

---

## Round 3 · List of Changes

We added Sections 3.4 and 4.6. We significantly revised Section 4.5 and the second part of Section 2.1. We added a Table at the end of Section 3.3. We gave a proper credit to Phys. Rev. B 102, 075310 (2020).

---

## Editorial Decision

published